# Label-Free SERS Analysis of Serum Using Ag NPs/Cellulose Nanocrystal/Graphene Oxide Nanocomposite Film Substrate in Screening Colon Cancer

**DOI:** 10.3390/nano13020334

**Published:** 2023-01-13

**Authors:** Jie Li, Qiutian She, Wenxi Wang, Ru Liu, Ruiyun You, Yaling Wu, Jingzheng Weng, Yunzhen Liu, Yudong Lu

**Affiliations:** 1Fujian Provincial Key Laboratory of Advanced Oriented Chemical Engineer, Fujian Key Laboratory of Polymer Materials, College of Chemistry and Materials Science, Fujian Normal University, Fuzhou 350007, China; 2College of Materials and Chemical Engineering, Institute of Oceanography Minjiang University, Fuzhou 350108, China

**Keywords:** graphene oxide, label-free, PCA-LDA, colon cancer, SERS

## Abstract

Label-free surface-enhanced Raman scattering (SERS) analysis shows tremendous potential for the early diagnosis and screening of colon cancer, owing to the advantage of being noninvasive and sensitive. As a clinical diagnostic tool, however, the reproducibility of analytical methods is a priority. Herein, we successfully fabricated Ag NPs/cellulose nanocrystals/graphene oxide (Ag NPs/CNC/GO) nanocomposite film as a uniform SERS active substrate for label-free SERS analysis of clinical serum. The Ag NPs/CNC/GO suspensions by self-assembling GO into CNC solution through in-situ reduction method. Furthermore, we spin-coated the prepared suspensions on the bacterial cellulose membrane (BCM) to form Ag NPs/CNC/GO nanocomposite film. The nanofilm showed excellent sensitivity (LOD = 30 nM) and uniformity (RSD = 14.2%) for Nile Blue A detection. With a proof-of-concept demonstration for the label-free analysis of serum, the nanofilm combined with the principal component analysis-linear discriminant analysis (PCA-LDA) model can be effectively employed for colon cancer screening. The results showed that our model had an overall prediction accuracy of 84.1% for colon cancer (*n* = 28) and the normal (*n* = 28), and the specificity and sensitivity were 89.3% and 71.4%, respectively. This study indicated that label-free serum SERS analysis based on Ag NPs/CNC/GO nanocomposite film combined with machine learning holds promise for the early diagnosis of colon cancer.

## 1. Introduction

Screening blood analysis methods for colon cancer can greatly enhance the detection of lesions in the population, improve patient prognosis, and substantially increase patient survival [1]. Surface-enhanced Raman scattering (SERS) is an optical sensing method that provides information on the molecular structural characteristics and the composition of the substance being detected. Therefore, the SERS technology which can identify and recognize structural differences between cancer and normal tissue is expected to furnish the diagnosis of colon cancer using blood analysis [2,3]. Unfortunately, the SERS spectrum of blood was complex, with minor differences in the spectrum between normal and cancer patients. Shangyuan Feng et al. [4] proposed a simple label-free blood test, which investigated the feasibility of silver nanoparticles (Ag NPs) colloids as SERS substrates and applied multivariate statistical techniques to assess the plasma differences between cervical cancer patients and healthy populations to achieve the nondestructive detection of cervical cancer. However, the reproducibility of the sample data was limited by the instability and inhomogeneity of the Ag NPs substrate.

Traditional SERS substrates generally employ hard base materials to enhance their stability such as silicon wafers, glass, quartz, and aluminum. These types of matrices are mass-heavy, high cost, low flexibility, and have poor biocompatibility, which limits their application [5,6,7]. Cellulose nanocrystals (CNC) are needle or rod-shaped compounds, and the properties of cheap, biodegradable, and low Raman background make CNC a promising SERS substrate material [8]. Apart from this, it also has unique morphological and optical properties, which make CNC easy to load or assemble metal nanoparticles by acting as a stable carrier platform in the SERS substrate design [9]. Qing Zhang achieved the controlled loading of gold nanospheres (Au NSs) on CNC by electrostatic adsorption self-assembly [10]. Segun A. Ogundare used CNC as a reducing agent and stabilizer during the formation of Ag NPs to synthesize effective SERS substrates [11]. However, the “coffee ring effect” would appear on these substrates after drying and before SERS measurement, which can lead to possible SERS signal fluctuations and poor signal reproducibility. As a result, the “solution-based” substrate needs further optimization for blood testing [12,13].

Graphene oxide (GO) allowed the formation of a homogeneous coating on the solid surface due to the superior biocompatibility and chemical inertness provided by the reactive oxygen sites, which was able to improve the reproducibility of the SERS technique. GO also maintained its exceptional surface properties, which made it easy to assemble in water. Thus, GO is expected to be adapted as a flexible material in the field of blood analysis [14,15].

In this study, a novel nanocomposite film was constructed by assembling CNC onto a small amount of GO and spin-coating by taking advantage of GO’s ease of dispersion, film formation, and enhanced adsorption of molecular signals. This nanocomposite film has improved the reproducibility of Ag NPs/CNC and eliminated the signal inhomogeneity that can cause deviations in blood analysis. As shown in Figure 1, we dropped serum samples on the Ag NPs/cellulose nanocrystal/graphene (Ag NPs/CNC/GO) nanocomposite film, dried them, and then performed SERS measurements directly. Finally, we trained the classification of various serum spectra using the PCA-LDA model. That is the first system to detect colon cancer using novel carbon nanocomposite film combined with machine learning algorithms for label-free screening. This work is expected to provide the potential for the application of carbon nanomaterials in biomedicine.

## 2. Materials and Methods

### 2.1. Materials

Silver nitrate (AgNO_3_ > 99.0%) and sodium borohydride (NaBH_4_ > 98.0%) were purchased from Sinopharm Chemical Reagent Co., Ltd. (Shanghai, China). Cellulose nanocrystals (CNC) were supplied by Macklin Biochemical Technology Co., Ltd. (Shanghai, China). Graphene oxide (GO = 2 mg/mL) solution was obtained from Suzhou Tanfeng Graphene Technology Co., Ltd. (Suzhou, China). Nile blue A (NBA) was purchased from Aladdin Bio-Chem Technology Co., Ltd. (Shanghai, China). All reagents were used without further purification and ultrapure water served as a dispersant. Bacterial cellulose membrane (BCM) was purchased from Guilin Qhong Technology Co., Ltd. (Guilin, China).

Human serum (normal, colon cancer) was provided by the Department of Oncology, Fuzhou Provincial Hospital, Fujian Province, China. The hospital has informed volunteers and patients and obtained their written consent.

### 2.2. Synthesis of Ag NPs/CNC Substrates and Ag NPs/CNC/GO Suspensions

Ag NPs/CNC substrates were prepared by improving on the method of Liu et al. [16]. Specifically, 1 mL of 10 mM AgNO_3_ solution was mixed with 1.5 mL of 1 wt% CNC water suspension and subjected to vigorous stirring for 1 h. The same concentration (10 mM) of different volumes (0 mL, 0.2 mL, 0.4 mL, 0.6 mL, 0.8 mL, 1.0 mL and 2.0 mL) of NaBH_4_ solution was then added dropwise into the CNC and AgNO_3_ suspensions with magnetic stirring at room temperature to prepare Ag NPs/CNC substrates.

Ag NPs/CNC/GO suspensions were prepared based on the above-mentioned improvement. Figure 1 showed the synthesis process of Ag NPs/CNC/GO materials. 1.5 mL of CNC dispersion and 10 μL of GO dispersion were mixed thoroughly under magnetic stirring for 30 min, and then adding 1 mL of 10 mM AgNO_3_ solution and 0.8 mL of 10 mM NaBH_4_ to the mixture with continuous stirring for 2 h to prepare the suspensions.

### 2.3. Preparation of Ag NPs/CNC/GO Nanocomposite Film

The Ag NPs/CNC/GO suspensions were centrifuged at 11,000 rpm for 30 min, and the bottom solids were collected and redispersed in ultrapure water to thoroughly clean them. Finally, the nanosuspensions were slowly and uniformly spin-coated onto the bacterial cellulose membrane (BCM) by the spin-coating method. Where the BCM was sonicated in ultrapure water and anhydrous ethanol for pretreatment in order to clean it well, after which it was dried in a blast oven at 50 °C to serve as a support layer for the suspensions. Eventually, the Ag NPs/CNC/GO nanocomposite film was prepared.

### 2.4. SERS Measurement on Nanocomposite Film

The SERS spectra were collected using a HORIBA Raman analyzer (XPLORA plus) from Kyoto, Japan, where the serum samples were excited using a 532 nm laser, and the 785 nm laser was used to characterize the SERS properties of the film. In this study, all SERS spectra were integrated for 10 s at 15 mW laser power using a 50× objective (NA = 0.75), and a 520 cm^−1^ single crystal silicon peak was selected for calibration. Where the spot diameter (d = 1.22 λ/NA) and the spatial resolution (r = 0.61 λ/NA) of the 785 nm laser were 1277 nm and 638 nm, respectively, and the spot diameter and the spatial resolution of the 532 nm laser were 865 nm and 433 nm. The membranes are highly hydrophilic (the reference Appendix A for the characterization of hydrophilicity), and to avoid the loss of Ag NPs on the Ag NPs/CNC/GO nanofilm, we spotted the NBA and serum samples directly on the film substrates, and the SERS measurements were performed directly after the samples were completely dried.

### 2.5. SERS Data Processing and Analysis

The raw spectra obtained from the experiment included Raman scattering, autofluorescence and noise signals, so we used Lab-spec 6 software (Horiba Scientific, Kyoto, Japan) to fit all spectra to the background curve as a baseline for background derivation. To compensate for the overall difference in spectral response due to physical effects, the spectral data in the range of 200–1800 cm^−1^ were area normalized. Furthermore, we used principal component analysis (PCA) to downscale the original SERS data, using the values of several principal components (PCs) instead of the numerous variables of the original spectra. Subsequently, a *t*-test has been used to find the most diagnostically significant PCs, and finally, these PC values have been imported into the linear discriminant analysis (LDA) for discriminant diagnostic analysis. The LDA process will use ‘leave-one-out cross-validation’ to discriminate the spectral data. Lastly, a subject operating characteristic curve (ROC) was introduced to further evaluate the diagnostic efficacy of the serum SERS-based technique for colon cancer.

### 2.6. Characterizations

Transmission electron microscopy (TEM) was performed using an FEI Tecnai 12 microscope operating at 200 kV (FEI Corp, Hillsboro, Oregon, USA). Scanning electron microscopy (SEM) (Sigma 300, ZEISS, Oberkochen, Germany) was used to characterize the surface and cross-sectional morphology of the nanocomposite films. UV-visible absorption spectra were recorded at room temperature by a UV 1902 spectrophotometer (Cold Light Technology, Shanghai, China) using a 600 μL black quartz cuvette. The size distribution and Zeta potential of nanoparticles of suspensions were determined using Zeta-sizer Nano ZS (Malvern Panalytical, Malvern, UK). The IR spectra were measured using an Attenuated Total Reflectance Fourier Transform Infrared (ATR-FTIR) instrument from Thermo Fisher Scientific (Waltham, MA, USA), model Nicolet iS 50. X-ray photoelectron spectroscopy (XPS) equipped with a flood gun (ESCALAB Xi+, Thermo Fisher Scientific) was performed to analyze the chemical composition of the films.

## 3. Results and Discussion

### 3.1. Characterization of Ag NPs/CNC/GO Suspensions

Figure 2A showed the color change of the solutions involved in the preparation of Ag NPs/CNC/GO suspensions. CNC, as a needle-like compound (in Figure 2E), was easily dispersed in water and appeared milky white. We used the method provided by Liu et al. [16] to prepare Ag NPs/CNC suspensions. When mixtures of CNC and AgNO_3_ reacted with NaBH_4_ solution, colorless Ag^+^ on the suspensions turned orange-yellow upon reduction, indicating the formation of Ag NPs. The optical properties of the Ag NPs/CNC suspensions obtained by UV-Vis absorption spectrometry were determined in Figure 2B. Significant plasmon resonance peaks were not observed in the absorption spectra of CNC dispersions. In contrast to CNC, the Ag NPs/CNC showed a strong surface plasmon resonance band at around 418 nm. In order to obtain Ag NPs/CNC suspensions with optimal SERS enhancement, we performed optimization by controlling the volume of NaBH_4_. It was found that the plasmon resonance absorption peak of the generated Ag NPs/CNC suspensions experienced a blue shift from 420 nm to 407 nm as the amount of NaBH_4_ was increased from 0.2 mL to 2.0 mL, as shown in Appendix A. Nile blue A (NBA) has been used as a detection molecule to characterize the Raman scattering enhancement of nanomaterials. NBA is dominated by a characteristic peak at 593 cm^−1^, which is attributed to the collective ring respiration pattern of the structure [17]. As in Appendix A, the best SERS signal was found for 0.8 mL NaBH_4_, especially at the peak of 593 cm^−1^, so we considered the amount at this point as the best result for the preparation of Ag NPs/CNC suspensions. However, Ag NPs appeared to agglomerate in the area of CNC stacking, as shown in the yellow area in Figure 2G.

To improve the uniformity of Ag NPs/CNC, we used GO for adjustment. The GO dispersion was a brownish-yellow nanomaterial with a folded transparent layer-like structure on its surface [18], as shown in Figure 2F. GO generally has a distinctive characteristic absorption peak at 231 nm, which is caused by the *π-π* transition absorption of the C=C bond of the aromatic ring, and a very weak absorption peak at 301 nm, as shown in Figure 2B. GO can provide loading sites for Ag NPs because of its high specific surface area and abundant active functional groups on the surface. As shown in the UV-Vis absorption and SERS spectra of Appendix A, we explored the content of GO dispersions (0.01 mL, 0.1 mL, 0.5 mL, 1 mL, 2 mL) added to the Ag NPs/CNC suspensions. With the increasing volume of GO, the characteristic peak \of GO UV-Vis absorption shifted to the short-wave direction from 230 nm to 207 nm. It’s indicated that GO gradually became smaller, from a medium size to a small size, which was mainly attributed to the partial reduction in GO, structural breakage, and size reduction [19,20]. With the gradual disappearance of the Raman peak of NBA, the Raman peak of GO became stronger and stronger, particularly in the peak of D centered at 1316 cm^−1^ and the peak of G centered at 1598 cm^−1^, which were the most prominent [21]. Therefore, we finally took the decision to add GO of 0.01 mL.

Eventually, the Ag NPs/CNC/GO suspensions exhibited a wider UV-Vis absorption plasmon resonance band. 417 nm is the absorption peak of Ag NPs and 215 nm is the plasmon resonance absorption peak of GO. To monitor the synthesis process, changes in surface charge were determined at each stage of the suspension fabrication by Zeta potential measurements, as shown in Figure 2C. The GO exhibited negative electrical properties at high concentrations. When the mixture of CNC and GO was reduced on Ag NPs, the Zeta potential was between −35.3 mV (CNC) and −49.2 mV (GO), which suggested that the Ag NPs/CNC/GO had been generated. As Figure 2D, the size distribution of CNC dispersion was around 122.4 nm, and yet the size of Ag NPs/CNC and Ag NPs/CNC/GO suspensions shifted to the right, which further verified that the presence of Ag NPs increased the size of the suspensions. We added a trace amount of GO to the CNC suspension and the Ag NPs were uniformly dispersed on the CNC and GO. The layered structure of GO provided a uniform dispersion of the Ag NPs loaded CNC, in the red position in Figure 2H. Therefore, we can say that the addition of GO dispersion improves the agglomeration phenomenon of Ag NPs/CNC suspensions. The inset has shown the distribution of Ag NPs in the Ag NPs/CNC/GO suspensions under magnification. The results demonstrated that Ag NPs were densely distributed in the CNC and GO suspension, with average particle sizes of 23.64 nm and 22.37 nm, respectively. (The particle size distributions were calculated by Nano Measurer 1.2 software. (Jie Xu, Laboratory of Surface Chemistry and Catalysis, Department of Chemistry, Fudan University, Shanghai, China. 

### 3.2. Characterization of Ag NPs/CNC/GO Nanocomposite Film

We prepared Ag NPs/CNC/GO nanocomposite film by homogeneously coating the suspensions onto BCM using a spin-coating method. In contrast to other cellulose membranes, the BCM exhibited a dense networking structure (Appendix A) that prevented CNC and Ag NPs from penetrating the membrane pore and reduced the SERS signal. The ATR-FTIR of Ag NPs/CNC/GO nanocomposite film was shown in Figure 3A. The broad absorption band centered at 3362 cm^−1^ for GO-BCM and Ag NPs/CNC/GO film was attributed to the O-H bond stretching vibrations from the -OH and -COOH groups. In contrast to GO-BCM, the characteristic peak of the C=O bond near 1738 cm^−1^ of the Ag NPs/CNC/GO composite film was dramatically weakened, indicating that the GO in the complex film was reduced [22]. The Ag NPs/CNC-BCM showed no apparent absorption peaks compared with CNC-BCM, only a small enhancement of the peak intensity. It is attributed to the fact that the interaction between CNC and Ag NPs is caused. The variation between Ag NPs/CNC/GO nanocomposite film and Ag NPs/CNC-BCM can be attributed to the effect of GO on the complexes we synthesized.

The relative numbers of C, O, and Ag atoms of the films were confirmed by XPS analysis, as shown in Figure 3B. As expected, the XPS spectra of Ag NPs/CNC-BCM and Ag NPs/CNC/GO nanocomposite film detected three distinct peaks for the Ag 3p, Ag 3d, Ag 4p energy level orbitals, which were the peaks attributed to Ag NPs. It is shown in Figure 3C that the Ag 3d track had two prominent peaks at 374.3 eV and 368.2 eV, which were designated as Ag 3d_3/2_ and Ag 3d_5/2_. Compared with CNC-BCM, Ag NPs/CNC/GO film exhibited a higher intensity of O1s (533.4 eV) orbitals due to the contribution of GO to the composite film with increased oxygen content. In Figure 3D, the carbon spectra of the C1s orbitals of CNC-BCM, GO-BCM, Ag NPs/CNC-BCM, and Ag NPs/CNC/GO nanocomposite films can be resolved into three types of groups, which were C1 based on the C-C bond and C-H bond, C2 based on the C-O bond, and C3 based on the C=O bond. The manifestation peak was about 284.8 eV for the C1 group, whose relative content of GO-BCM was remarkably greater than that of the other membranes. The C2 groups that exhibited the 286.2 eV~286.8 eV region showed a deviation of 0.3 eV from the peak positions of the four films. By comparison, the C3 group with 287.8 eV as the expression peak has the smallest number in CNC-BCM, which showed similar results to the literature [23,24,25].

Figure 3E demonstrated the morphological structure of the Ag NPs/CNC/GO nanocomposite films. We saw that the film exhibited a smooth surface, which was mainly attributed to the dispersing effect of trace GO on the CNC. To obtain a good enhancement of the substrate, we further adopted multiple spin-coating in small doses during the film-making process, thus the nanomembrane displayed a multilayer structure, as in Figure 3F, which provided us with sufficient SERS “hot spots”. For a clearer knowledge of the Ag NPs content, we have tested the content of three synthesised Ag NPs/CNC/GO nanofilms in parallel using ICP-OES; the results are listed in Appendix A. We achieved up to 0.1 mg/cm^2^ of silver on the Ag NPs/CNC/GO nanocomposite film.

### 3.3. Sensitivity and Uniformity of Ag NPs/CNC/GO Nanocomposite Film

In the wavelength range of 400–1800 cm^−1^, CNC-BCM possesses a very low Raman background peak, but GO-BCM exhibits strong D and G peaks, which have an impact on the signal of NBA. Therefore, we successfully prepared nanofilm using trace amounts of GO as an additive to homogenize the signals of Ag NPs/CNC-BCM without affecting the main NBA peaks, as shown in Figure 4A. The peak at 593 cm^−1^ was left as a quantitative peak to assess the SERS performance. The Ag NPs/CNC/GO nanocomposite film adsorbed with NBA showed a slight bump in peak shape at 1316 cm^−1^ and 1598 cm^−1^, owing to the presence of GO; still, the peak at wavelength 593 cm^−1^ showed a unique and powerful signal. In Figure 4B, we characterized the sensitivity of the nanocomposite film. The SERS intensity was reduced as the concentration of NBA decreased from 10^−4^ M to 10^−8^ M (illustrated in the inset in Figure 4B), and the 593 cm^−1^ SERS intensity and concentration showed a linear correlation with the linear correlation coefficient (R^2^) of 0.9932. Taking the limit of detection (LOD) as three times the standard deviation of the blank, the detection limit of the Ag NPs/CNC/GO nanocomposite film for NBA could reach 30 nM. To better highlight the SERS performance of our nanocomposite film, we further characterized Ag NPs/CNC-BCM using NBA under the same conditions, as shown in Appendix A. The SERS intensity and concentration were linearly correlated, but the R^2^ (0.9084) was low, which was attributed to the stacking between CNC molecules and the interaction with Ag NPs [12]. In Figure 4C,D and Appendix A, we further verified that the addition of GO can reduce the agglomeration of Ag NPs/CNC-BCM by comparing the SERS mapping array and SERS waterfall plots of both Ag NPs/CNC/GO nanocomposite film (relative standard deviation, RSD = 14.2%) and Ag NPs/CNC-BCM (RSD = 25.3%) substrates.

### 3.4. Label-Free Detection of Colon Cancer Serum Samples

The serum molecule is an organic substance, and when the laser is irradiated for a long time, there is some damage, so we discussed the laser power and laser irradiation time. Based on the better analysis speed and accuracy, a 532 nm laser was chosen here for the measurement of serum samples. Figure 5A, B showed the SERS spectra obtained by taking a case of serum from the colon cancer group and the normal group adsorbed on the nanocomposite film under the laser power of 15 mW, irradiated by the laser at the same position for 10 s and tested 6 times in parallel. We observed that the SERS spectra of the serum emerged at the same peak position, and the SERS intensity remained at the maintenance level. Therefore, the laser conditions we chose have no loss of SERS spectra of serum molecules within the total irradiation time of 60 s. To further verify the above conditions, we also measured the damage to the serum by controlling different irradiation times (3 s, 5 s, 7 s, 10 s, 12 s). As shown in Figure 5C–F, we saw that the peak intensity of the serum SERS spectra of the colon cancer group and the normal group were at the same peak position with an increasing integration time while keeping the laser power of 15 mW constant, with the characteristic peak of 638 cm^−1^ as the reference peak, which gradually increased. Considering the complexity of the serum spectral peaks and the concise operation of the experiment, we finally chose the 532 nm laser irradiation for 10 s at a laser power of 15 mW for SERS measurement of serum.

We first collected SERS spectra of different serum samples using Ag NPs/CNC/GO nanocomposite film, in which samples from 28 colon cancer groups and 28 normal groups were collected, and 6 SERS spectra were collected in parallel for each sample, and the average spectrum was used as the metadata for label-free analysis. As illustrated in Appendix A, we presented the RSD values of the nanomembranes for serum from the colon cancer and normal groups at their characteristic peaks of 638 and 1139 cm^−1^, which we calculated to correspond to the mean RSD of 9.05% (Appendix A) and 10.94% (Appendix A) for the colon cancer and normal groups, respectively; both values are lower than the classical value of the SERS bases. Therefore, our Ag NPs/CNC/GO nanocomposite film also showed excellent reproducibility for the detection of serum.

Figure 6A showed the average SERS spectra of our test colon cancer groups and normal groups as well as the difference spectra, displaying essentially similar positions of the spectral peaks for both serum groups, such as 494, 595, 636, 726, 810, 889, 1014, 1101, 1135, 1208, 1334, 1452, 1582, and 1685 cm^−1^. This can be tentatively assigned to specific biomolecules closely related to DNA/RNA, proteins, sugars, and lipids based on the published literature [26,27], as shown in Appendix A. The purine band still dominated the spectra, but there were also some significant differences between the two SERS spectra [28,29]. Owing to the different levels of various biological components in the serum, such as amino acids, collagen, amides, etc., the intensity of the corresponding SERS signal was also affected. In this study, we have counted the differences in the intensity of the specific spectral peaks depicted in Figure 6B. The peaks marked in yellow in the bar chart, 636, 726, 1334, and 1582 cm^−1^, have higher SERS intensity in the colon cancer group than in the normal group, resulting in increased levels of tyrosine, hypoxanthine, collagen, and phenylalanine in the serum of the cancer compared with the normal group. This was attributed to a mutation in the colon cancer group that reduced the activity of enzymes such as phenylalanine hydrogenase, which decreased the rate of protein degradation, allowing phenylalanine and its metabolites to accumulate in the body and cause disease, as has been successfully demonstrated in cancer research. Conversely, SERS bands at 494, 595, 810, 889, 1012, 1135 and 1208 cm^−1^ were less intense in the colon cancer group than in the normal group, mainly explained by abnormal base metabolism in colon cancer patients, resulting in reduced amino acid and sugar content [30,31,32,33].

We conducted PCA-LDA on the total SERS spectra using the SPSS 19.0 software package (SPSS Inc., Chicago, IL, USA) to assess the serum SERS spectra and its ability to screen the colon cancer and normal samples [34]. Figure 6C has shown 2D scatter plots obtained by selecting two columns of diagnostically significant PCs (PCs < 0.005; PC1 = 0.001, PC2 = 0.003) scores, and the sensitivity and specificity of our test method were calculated, combining the discriminant scatter plot of posterior probabilities in Figure 6D. Figure 6E demonstrated the receiver operating the characteristic curves (ROC) for sensitivity and specificity. The area under the curve (AUC) represented the predictive accuracy of the classification, with AUC closer to 1.0, implying a better diagnostic test. Furthermore, the results of the *t*-test analysis of the independent samples were used to distinguish between the colon cancer and normal groups. It is concluded that our model has an overall prediction accuracy of 84.1%, with a model specificity of 89.3% and a sensitivity of 71.4%; the confusion matrix of prediction results in Figure 6F. This is to say that 8 of the 28 spectra from the serum samples in the colon cancer groups were misclassified as controls and only 3 of the 28 spectra from the serum samples in the normal groups were misclassified as colon cancer. Thus, the differences in the SERS spectra of the colon cancer and normal groups and the misclassification of the PCA-LDA model indicated abnormal metabolic profiles and changes in the composition of serum proteins in the colon cancer groups [33,35].

## 4. Conclusions

In this work, we developed Ag NPs/CNC/GO nanocomposite film that showed excellent reproducibility in detecting NBA and serum and also effectively improved the disadvantage of the Ag NPs/CNC substrate, which tends to agglomerate. Most importantly, we presented a label-free serum SERS analysis method using the Ag NPs/CNC/GO nanocomposite film to distinguish between colon cancer patients and normal individuals. This method combined nanomaterials and machine learning to enable label-free SERS detection for colon cancer and also has potential applications for screening other cancers, such as gastric and lung cancers.

## Figures and Tables

**Figure 1 nanomaterials-13-00334-f001:**
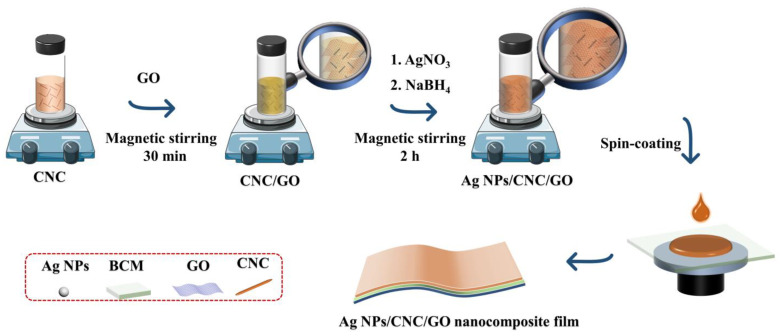
Diagram of the preparation process of Ag NPs/CNC/GO nanocomposite films using self-assembly.

**Figure 2 nanomaterials-13-00334-f002:**
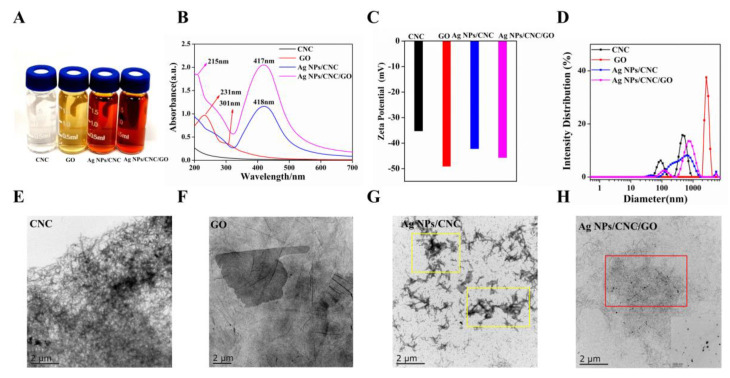
Characterization of CNC, GO, Ag NPs/CNC and Ag NPs/CNC/GO suspensions. (**A**) Optical images (from left to right). (**B**) UV-Vis absorption spectra. (**C**,**D**) Zeta potential of the suspensions and the corresponding size distribution changes. (**E**–**H**) respectively showed the respective TEM images, where the inset in (**H**) was a corresponding enlarged image.

**Figure 3 nanomaterials-13-00334-f003:**
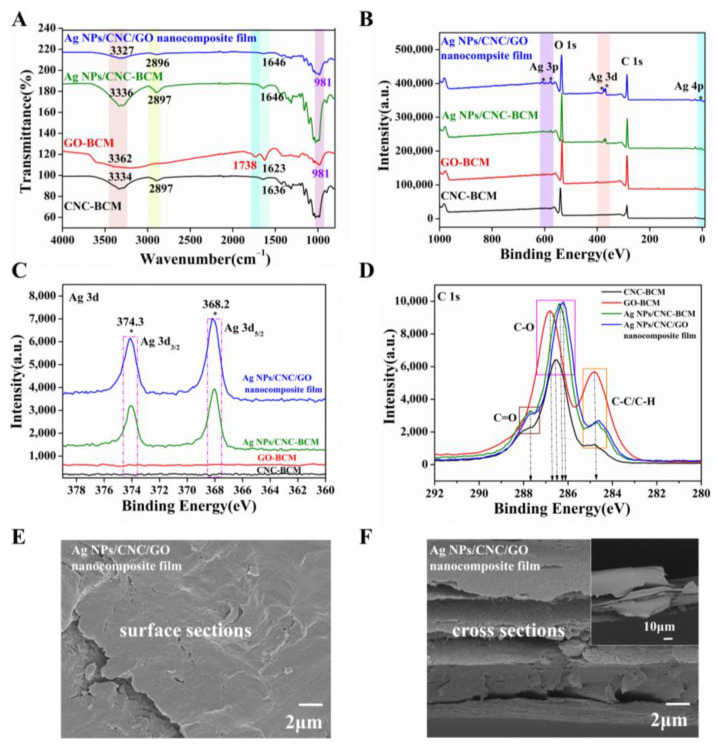
We spin-coated CNC, GO, Ag NPs/CNC and Ag NPs/CNC/GO suspensions onto BCM (serially named as CNC-BCM, GO-BCM, Ag NPs/CNC-BCM, Ag NPs/CNC/GO nanocomposite film). (**A**–**D**) were the spectra of CNC-BCM, GO-BCM, Ag NPs/CNC-BCM and Ag NPs/CNC/GO nanocomposite film, (**A**) ATR-FTIR spectrum; (**B**) XPS full spectrum; (**C**) XPS partial spectrum of Ag 3d; (**D**) XPS partial spectrum of C1s. (**E**,**F**) the surface and cross-sectional SEM images of the Ag NPs/CNC/GO nanocomposite film, where the inset in (**F**) was a corresponding large size.

**Figure 4 nanomaterials-13-00334-f004:**
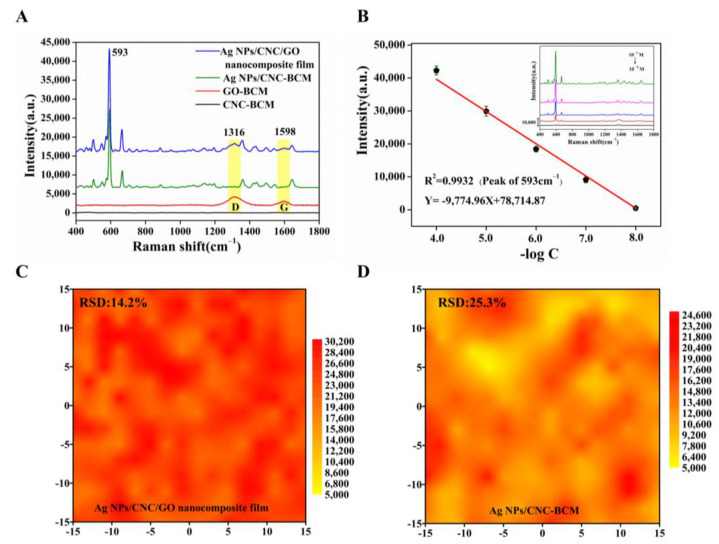
We characterized the SERS performance of our synthesized Ag NPs/CNC/GO nanocomposite film using a 785 nm laser integrated at 15 mW laser power for 10 s. (**A**) SERS spectra of NBA (10^−5^ M) adsorbed on CNC-BCM, GO-BCM, Ag NPs/CNC-BCM and Ag NPs/CNC/GO nanocomposite film. (**B**) Linear curve of 593 cm^−1^ SERS intensity versus concentration for NBA adsorbed on Ag NPs/CNC/GO nanocomposite film, inset indicates SERS spectra of NBA at 10^−4^ M–10^−8^ M. (**C**,**D**) were the SERS mapping plots at 593 cm^−1^ of the SERS spectra of Ag NPs/CNC/GO nanocomposite film and Ag NPs/CNC-BCM in 30 μm × 30 μm arrays with 10^−5^ M NBA adsorbed, respectively.

**Figure 5 nanomaterials-13-00334-f005:**
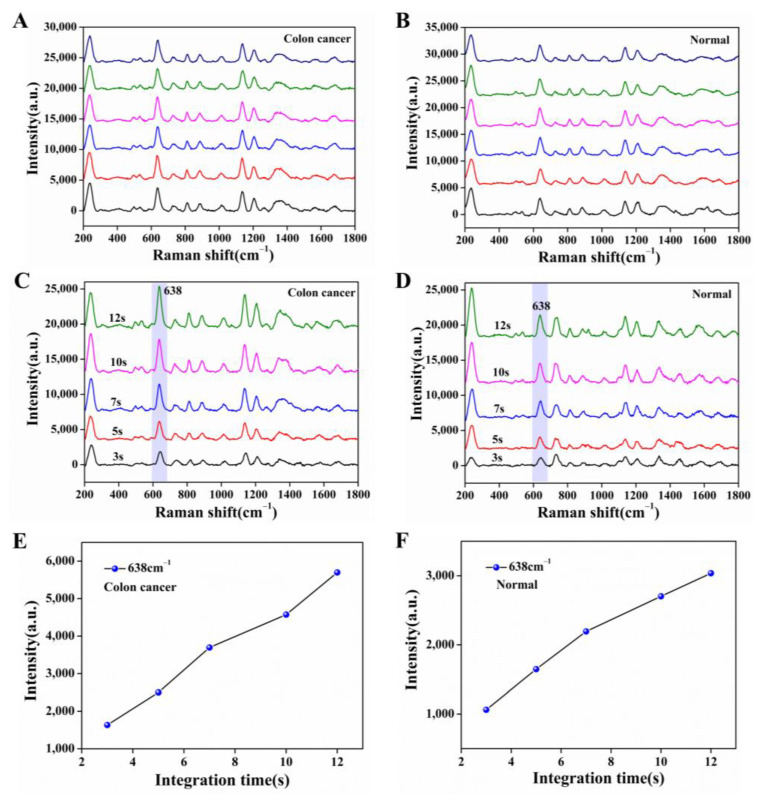
SERS spectra of serum adsorbed on Ag NPs/CNC/GO nanocomposite film at the same site. (**A**,**B**) respectively represented the colon cancer group and the normal group tested six times in parallel under the same laser conditions (laser power (15 mW), irradiation time (10 s); (**C**,**D**) were maintained the laser power (15 mW) was constant and only the irradiation time (3 s, 5 s, 7 s, 10 s, 12 s) was changed for the colon cancer group and normal group. (**E**,**F**) were corresponding to the SERS intensity of the SERS spectra of (**C**,**D**) at the characteristic peak 638 cm^−1^, respectively.

**Figure 6 nanomaterials-13-00334-f006:**
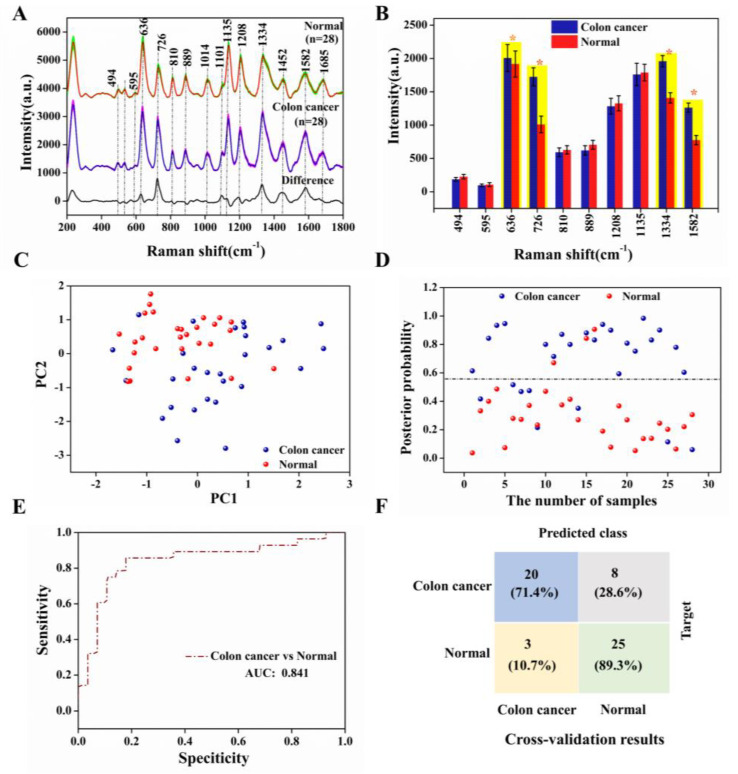
(**A**) Averaged SERS spectra of the colon cancer (*n* = 28) and normal (*n* = 28) serum samples and the difference spectra, where the shaded areas represent the mean standard deviations; (**B**) The corresponding histograms of the average intensities and standard deviations of the SERS peaks of the colon cancer and normal groups; (**C**) Two-dimensional scatter plot using two PCs for the colon cancer and normal groups; (**D**) The discrimination scatter plot scores of the posterior probabilities using two PCs for the colon cancer and normal groups; (**E**) ROC classification results for the PCA-LDA analysis generated groups using the leave-one-out and cross-validation methods (AUC: integration areas under the ROC curves); (**F**) Confusion matrices of the colon cancer and normal groups prediction results using PCA-LDA analysis.

## Data Availability

The data presented in this study are available on request from the corresponding author.

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
