# Peer review of "Label-Free SERS Analysis of Serum Using Ag NPs/Cellulose Nanocrystal/Graphene Oxide Nanocomposite Film Substrate in Screening Colon Cancer"

_nanomaterials, 2023, doi:10.3390/nano13020334_

Round 1
Reviewer 1 Report
In this paper, the authors fabricated AgNPs cellulose/nanocrystals/graphene oxide nanocomposite film (AgNP/CNC/GO) as a uniform SERS active substrate for label-free SERS analysis of clinical serum. Combined with machine learning analysis, the substrate holds promise for the early diagnosis of colon cancer. The uniformity and sensitivity of the AgNP/CNC/GO film is fair and the diagnosis looked successful. However, the paper is lacking many critical information for publishing. I recommend the author to rewrite the paper while checking detailed information including comments listed below.
(1) line116
Two kinds of laser 532 and 785nm is employed. Which wavelength is used for Figure4 and 5? What are the power density for SERS measurements ?
(2) The author should discuss the laser power density used in this experiment together with the sample and substrate damage, since the sample is organic molecules and not always safe for a laser irradiation.
(3) What kind of objective lens is used to obtain Figure 4C and D ?
(4) Line 306
What does it mean “PC<0.005 “ ?
(5) In PCA analysis, two components PC1 and PC2 are selected. What are the contribution of the two components with respect to total components ?
(6) According to figure 4 A, the spectra from the substrate little affects the SERS spectra of NBA, however the D-band component appeared at 1300 cm-1 will bother the analysis when applied to serum. How the authors deal with this back ground signal?
(7) Figure 5 F
Please check the numbers in the figure.
Author Response
Dear Reviewer
We are grateful for your consideration of our manuscript entitled “Label-free SERS analysis of serum using Ag NPs/ cellulose nanocrystal/ graphene oxide nanocomposite film substrate in screening colon cancer”(nanomaterials-2144276). We also appreciate the suggestions and comments made by all reviewers related to this manuscript, addressing which certainly helped to improve our work.
The following section of this document provides an itemized response to each of the reviewers' comments. We believe that addressing their concerns accordingly will improve the quality of our manuscript and hope that the revised manuscript will be of great interest to the scientific community in related fields.
Please note the following: black text corresponds to the reviewers' comments, the red text is our response to the reviewers' comments, and the red underlined text represents the changes in the revised manuscript.
In this paper, the authors fabricated Ag NPs cellulose/nanocrystals/graphene oxide nanocomposite film (Ag NPs/CNC/GO) as a uniform SERS active substrate for label-free SERS analysis of clinical serum. Combined with machine learning analysis, the substrate holds promise for the early diagnosis of colon cancer. The uniformity and sensitivity of the Ag NPs/CNC/GO film is fair and the diagnosis looked successful. . However, the paper is lacking many critical information for publishing. I recommend the author to rewrite the paper while checking detailed information including comments listed below.
Response: We thank the reviewer for appreciating our work. And thank you for reviewing our manuscript and for the constructive comments, which greatly helped us to improve the manuscript. We have revised the manuscript as you requested and thank you for your guidance.
(1) Line116
Two kinds of laser 532 and 785nm is employed. Which wavelength is used for Figure4 and 5? What are the power density for SERS measurements ?
Response: We appreciated the care and professionalism of the reviewers. In lines 116 to 125 of the manuscript, we fully supplement the laser power and integration time for the different laser wavelengths used (532 nm,785 nm). The wavelength and power of the laser used in Figures 4 and 5 have been added in the manuscript.
In this study, all SERS spectra were integrated for 10 s at 15 mW laser power using a 50x objective (NA=0.75), and 520 cm-1 single crystal silicon peak was selected for calibration. Where the spot diameter (d = 1.22 λ / NA) and the spatial resolution (r = 0.61 λ / NA) of the 785 nm laser were 1277 nm and 638 nm, respectively, and the spot diameter and the spatial resolution of the 532 nm laser were 865 nm and 433 nm.
Fig. 4 We characterized the SERS performance of our synthesized Ag NPs/CNC/GO nanocomposite film using a 785 nm laser integrated at 15 mW laser power for 10 s.
Fig. 5 We finally chose the 532 nm laser irradiation for 10 s at a laser power of 15 mW for SERS measurement of serum.
(2) The author should discuss the laser power density used in this experiment together with the sample and substrate damage, since the sample is organic molecules and not always safe for a laser irradiation.
Response: We appreciate the reviewer's suggestions. In the text of the manuscript we have added the effect of laser power density on our samples. (line 283-309)
Figure 5 SERS spectra of serum adsorbed on Ag NPs/CNC/GO nanocomposite film at the same site. A and B respectively represented the colon cancer group and the normal group tested six times in parallel under the same laser conditions (laser power (15 mW), irradiation time (10 s); C and D were maintained the laser power (15 mW) was constant and only the irradiation time (3 s, 5 s, 7 s, 10 s, 12 s) was changed for the colon cancer group and normal group. E and F were corresponding to the SERS intensity of the SERS spectra of C and D at the characteristic peak 638cm-1, respectively.
The serum molecule is an organic substance and when the laser is irradiated for a long time, there is some damage, so we discussed the laser power and laser irradiation time. Based on the better analysis speed and accuracy, a 532 nm laser was chosen here for the measurement of serum samples. Figs. 5A and 5B showed the SERS spectra obtained by taking a case of serum from colon cancer group and normal group adsorbed on the nanocomposite film under the laser power of 15 mW, irradiated by laser at the same position for 10 s and tested 6 times in parallel. We observed that the SERS spectra of the serum emerged at the same peak position and the SERS intensity remained the maintenance. Therefore, the laser conditions we chose have no loss of SERS spectra of serum molecules within the total irradiation time of 60 s. To further verify the above conditions, we also measured the damage to the serum by controlling different irradiation times (3 s, 5 s, 7 s, 10 s, 12 s). As shown in Fig. 5C-F, we saw that the peak intensity of the serum SERS spectra of the colon cancer group and the normal group were at the same peak position with increasing integration time while keeping the laser power of 15 mW constant, with the characteristic peak of 638 cm-1 as the reference peak, which gradually increased. Considering the complexity of the serum spectral peaks and the concise operation of the experiment, we finally chose the 532 nm laser irradiation for 10 s at a laser power of 15 mW for SERS measurement of serum.
(3) What kind of objective lens is used to obtain Figure 4C and D ?
Response: We thank the reviewer for reminding us. Figure 4C and D using a 50x objective lens. All spectra obtained in our work were measured under the 50x objective. At line 119, we also made additions.
(4) Line 306
What does it mean “PC<0.005”?
Response: Thank the reviewer for the concern. We conducted PCA-LDA on the total SERS spectra using the SPSS 19.0 software package (SPSS Inc, Chicago) to assess the serum SERS spectra and its ability to screen the colon cancer and normal samples. A t-test is involved in the calculation, which is used to calculate the magnitude of the difference between the numbers of the two groups (colon cancer group and normal group). If you measured the "p-value" of an experiment at 5%, that means you are 95% confident that the experiment is correct. In a formal experiment, the experiment is only considered to be OK if the "p-value" is less than 5%. The smaller the value, the more accurate the data and the higher the confidence level. PC scores represented the weight of a specific component relative to the fundamental spectrum and can reflect the differences between groups. Therefore, the "PC<0.005" here means that the scores we chose are significantly different.
(5) In PCA analysis, two components PC1 and PC2 are selected. What are the contribution of the two components with respect to total components ?
Response: Thank the reviewer for the concern. The percentages of the components of PC 1 and PC 2 that we chose for the total components were 17.8% and 9.4%.
(6) According to figure 4 A, the spectra from the substrate little affects the SERS spectra of NBA, however the D-band component appeared at 1300 cm-1 will bother the analysis when applied to serum. How the authors deal with this back ground signal?
Response: I apologize that this is a point we had not thought of and thank the reviewers for their insights. The characteristic peaks of GO are at 1316 and 1598cm-1. From the following serum SERS spectra, we basically do not see the effect of GO on the serum wave peaks. In addition to this, we tested the exact same conditions in the process of differentiating colon cancer and normal human serum spectra using the PCA-LDA algorithm, for example, both used Ag NPs/CNC/GO nanocomposite films for SERS measurements, which to some extent also attenuated the effect of the basal peaks on the spectra.
(7) Figure 5 F
Please check the numbers in the figure.
Response: We thank the reviewer for reminding us. We have checked the numbers in Figure 5F to make sure they are correct. Because this was the final result of our choice of the “leave-one-out cross-validation” method.

Reviewer 2 Report
The authors reported the use of surface-enhanced Raman scattering combined to PCA LDA analysis as a label-free technique for early diagnosis of colon cancer. They demonstrated that their model had an overall prediction accuracy of 84.1% for the colon cancer (n=28) and the normal (n=28), and the specificity and sensitivity were 89.3% and 71.4%, respectively. This study will be the first step before using this technique for screening other cancers, such as gastric and lung cancers
· The authors reported in the introduction that GO can effectively inhibit fluorescence interference in resonance spectroscopy measurements, which exactly overcome the defects of conventional SERS substrates. One of the advantages of SERS with a metallic nanoparticle surface is fluorescence quenching due to energy transfer between the fluorophore and the metal. This fluorescence quenching is not due to the presence of GO.
· The authors reported that for Ag NPs/CNC substrates synthesis, the same concentration of different volumes of NaBH4 solution was added dropwise into the CNC and AgNO3 suspensions with magnetic stirring. This concentration of NaBH4 is 10 mM of 1wt%. Same remark for Ag NPs/CNC/GO substrates
· Each spectrum was acquired with 1% laser power. The authors should precise the laser power on the sample. In addition, the authors should give the specifications of objective (NA, the diameter of laser spot) and the spectral resolution of the spectrometer.
· The authors should describe in more details the pretreatment method applied on spectra. They should also describe the instrument response correction, subtraction of interference (dark current, optics), and the spectral normalization.
· The authors reported in the material & methods section “Fourier transform infrared spectroscopy (FTIR), we characterized the functional groups on the films surface using a Nicolet 5700 from Thermo …”. Did the authors mean “Fourier transform infrared spectroscopy (FTIR), was used to characterize the functional groups on the films surface using a Nicolet 5700 from Thermo …”
· Figure S2C showed SERS spectra of Nile blue A on Ag NPs/CNC substrates synthesized by adjusting the volume of NaBH4 (0 mL, 0.2 mL, 0.4 mL, 0.6 mL, 0.8 mL, 1.0 mL and 2.0 mL). The authors reported that the best SERS signal was found for 0.8 mL NaBH4 while the absorbance spectrum of Ag NPs/CNC substrates synthesized with 0.6 mL was more intense to that synthesized with 0.8 mL.
· The authors explored the content of GO dispersions (0.01 mL, 0.1 mL, 0.5 mL, 1 mL, 2 mL) added to the Ag NPs/CNC suspensions (Fig. S2(d)). The intensity of the band at 593 cm-1 in the spectrum measured on Ag NPs/CNC substrates with 0.8 mL of NaBH4 (Fig. S2 C) than that with 0.01 mL of GO (Fig. S2 d). Can the authors better explain the role of GO on the enhancement properties of substrates?
· I did not understand that sentence in the results and discussion section “And the gradual disappearance of the Raman peak of NBA, the Raman peak of GO became stronger and stronger,…”
· In figure 5A the authors should add in this figure peak frequencies of the bands described in the results and discussion section. In addition, the authors reported that the most difference between these classes were attributed to the tyrosine, hypoxanthine, collagen, and phenylalanine. From my point of view, the Raman signal of collagen is not enhanced with the SERS method. The differences should be associated to the changes in nucleic acid content
· The authors should explain in more details the molecular differences between normal and cancer colon samples based on Raman bands. Additionally, authors should add an assignment table of all bands in the spectra shown in Figure 5A
Round 2
Reviewer 1 Report
The work is now sufficiently improved and ready to publish.
Reviewer 2 Report
Major comments were addressed by the reviewers to restructure the revised version of the manuscript. Almost of the reviewers’ suggestions have been taken into account. This article was improved and can be published in the journal